# Towards Higher Rate Electrochemical CO$_2$ Conversion: From Liquid-Phase to Gas-Phase Systems

**Jun Tae Song** [1,†], **Hakhyeon Song** [2,†], **Beomil Kim** [2] and **Jihun Oh** [1,2,3,*]

[1]  Graduate School of Energy, Environment, Water, and Sustainability (EEWS), Korea Advanced Institute of Science and Technology (KAIST), Daejeon 34141, Korea; sjunkun87@kaist.ac.kr
[2]  Department of Materials Science and Engineering, KAIST, Daejeon 34141, Korea;
hyeon0401@kaist.ac.kr (H.S.); beomil0907@kaist.ac.kr (B.K.)
[3]  KAIST Institute for NanoCentury (KINC), KAIST, Daejeon 34141, Korea
*   Correspondence: jihun.oh@kaist.ac.kr; Tel.: +82-042-350-1726
†   These authors contributed equally to this work.

**Abstract:** Electrochemical CO$_2$ conversion offers a promising route for value-added products such as formate, carbon monoxide, and hydrocarbons. As a result of the highly required overpotential for CO$_2$ reduction, researchers have extensively studied the development of catalyst materials in a typical H-type cell, utilizing a dissolved CO$_2$ reactant in the liquid phase. However, the low CO$_2$ solubility in an aqueous solution has critically limited productivity, thereby hindering its practical application. In efforts to realize commercially available CO$_2$ conversion, gas-phase reactor systems have recently attracted considerable attention. Although the achieved performance to date reflects a high feasibility, further development is still required in order for a well-established technology. Accordingly, this review aims to promote the further study of gas-phase systems for CO$_2$ reduction, by generally examining some previous approaches from liquid-phase to gas-phase systems. Finally, we outline major challenges, with significant lessons for practical CO$_2$ conversion systems.

**Keywords:** CO$_2$ reduction; catalysts; liquid-phase reactor; H-type cell; gas-phase reactor; membrane electrode assembly (MEA) cell; microfluidic cell; gas diffusion electrode (GDE); review

## 1. Introduction

Over the past century, the concentration of atmospheric carbon dioxide (CO$_2$) has steeply risen, and rising CO$_2$ levels are widely believed to be the main cause of global climate change. In response, substantial efforts have been devoted globally to not only deal with carbon emissions, but also to utilize CO$_2$ as a resource for beneficial processes. Among the methods utilizing CO$_2$, the electrochemical conversion of CO$_2$ has attracted considerable attention, for producing value-added fuels or chemicals while achieving a carbon neutral society. Notably, the required input energy for the electroreduction of CO$_2$ can be supplied by renewable energy. This offers a means of efficient storage of the intermittent renewable electricity generated by solar or wind, thereby providing greater reliability. In fact, there are numerous possible products (e.g., carbon monoxide (CO), formic acid (HCOOH), methane (CH$_4$), ethylene (C$_2$H$_4$), etc.) determined by various CO$_2$ reduction reaction (CO$_2$RR) pathways, with 2-, 4-, 6-, 8-, and 12-electron transfers, as shown in Figure 1 [1]. Major producible chemicals have industrial uses as precursors in chemical processes, fuels, and preservatives. In terms of the market size, ethylene and ethanol are highly desired products. Also, long-chain carbon products (more than C$_3$) are generally profitable because of their higher energy density [2]. However, the complexity of the CO$_2$RR pathway from CO$_2$ to the final product and the high number of required electrons make it difficult to obtain

these chemical products efficiently. Thus, it is imperative to develop efficient catalytic materials for $CO_2RR$ technology.

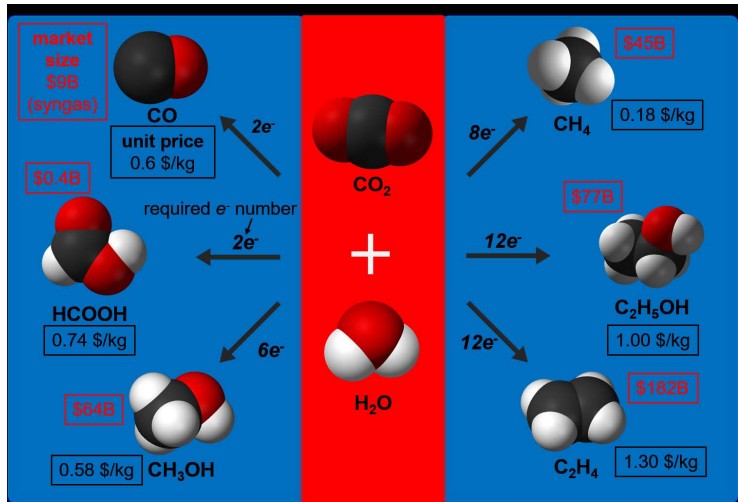

**Figure 1.** Various possible routes for the products from the $CO_2$ reduction reaction. The market size, unit price, and required number of electrons for producing each chemical are marked in the schematic [1].

Despite the necessity for such higher-order carbon products in industry, product selectivity is mostly determined by the intrinsic properties of the catalytic material, and as a result, only some products from the electrochemical $CO_2$ conversion are economically viable. Figure 2 shows the trends of Faradaic efficiencies for specific products from $CO_2RR$ by year [1]. As reported in a comprehensive work by Hori et al., as well as many other studies, $C_1$ products such as HCOOH and CO are easily obtainable utilizing Sn-, In-, Au-, and Ag-based catalysts with a considerably high Faradaic efficiency. In contrast, Cu-based materials are major catalysts producing $C_{2+}$ chemicals, but increasing the selectivity for $C_{2+}$ is difficult because of the various reaction pathway branches from $CO_2$ to the final product. In recent years, $C_{2+}$ selectivity has increased, as many researchers have focused on the development Cu catalysts to target higher-order chemicals, while research on $C_1$ to not only further improve selectivity, but also reduce the reaction overpotential, has also been consistently conducted. For the development of catalytic materials, researchers have sought to improve their performance with various strategies, such as nanostructuring, surface functionalization, alloying, and optimizing the electrolyte [3–18]. Meanwhile, many factors (e.g., morphology, grain boundary, size, shape, electronic structure, pH, etc.) affecting catalytic performance have been revealed, while mechanistic insight has also been provided via theoretical investigations [11,19–21].

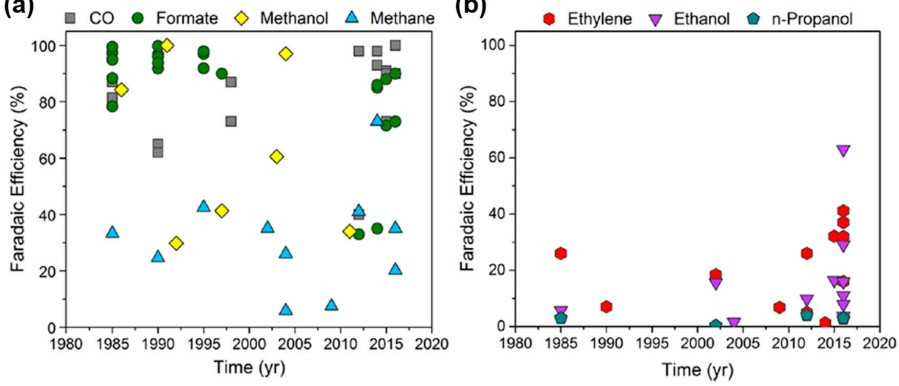

**Figure 2.** Faradaic efficiency records classified by year for (**a**) $C_1$ products and (**b**) $C_2$–$C_3$ products. Reprinted with permission from the authors of [1]. Copyright (2018) American Chemical Society.

Although there have been significant advances in $CO_2RR$ to date, the technology is still far from the level of practical utilization. A recent techno-economic analysis using a gross-margin model emphasizes a high-current density of over approximately 200 mA $cm^{-2}$, and the long-term durability for the economic feasibility of $CO_2RR$ technology [22]. The $CO_2RR$ electrolyzer systems are classified into two major categories, liquid-phase and gas-phase reactors, by the difference in the $CO_2$ reactant form, as shown in Figure 3. To date, the vast majority of previous work on $CO_2RR$ catalysts utilized a liquid-phase reactor, by supplying a $CO_2$ reactant to an electrolytic solution. This system, however, suffers from the mass transport limit because of the low solubility of $CO_2$ (33 mM), hindering high productivity. Thus, this liquid-phase-based cell configuration is not appropriate for commercially viability. In contrast to the liquid-phase reactor, gas-phase cell systems have recently shown an outstanding performance, in particular, recording a high current density for target chemical products, without regard for the mass transport limit of the reactant [17,18,23–29].

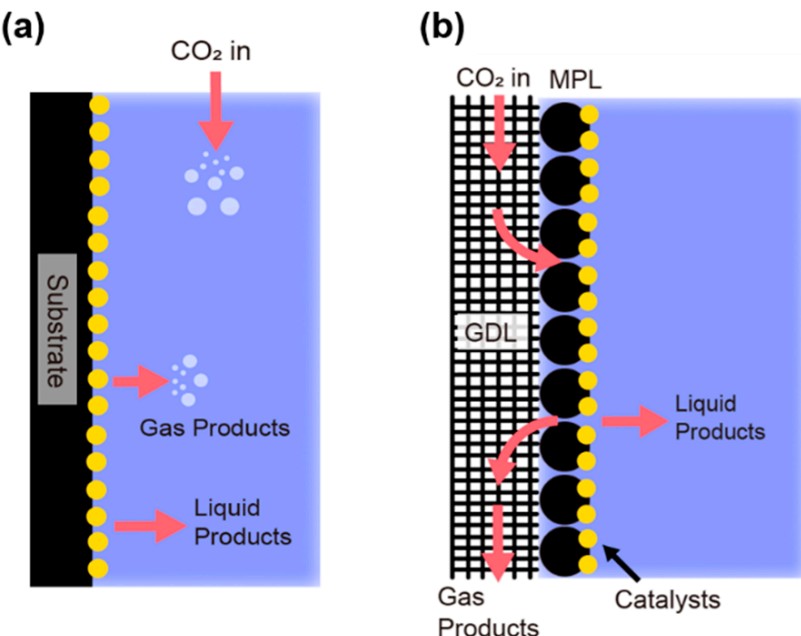

**Figure 3.** Schematics comparing the concept of (**a**) liquid-phase and (**b**) gas-phase $CO_2$ reduction reactions.

The general trend of the reported current densities for $CO_2RR$ ($j_{CO2 \ Reduction}$) is plotted in Figure 4. It clearly shows that gas-phase reactors for $CO_2$ reduction can achieve a higher current density over 100 mA $cm^{-2}$, while most records of $j_{CO2 \ Reduction}$ from a liquid-phase reaction are below 100 mA $cm^{-2}$. Accordingly, many researchers in this field have recently promoted the development of gas-phase reactors [18,23,24,27,30–32]. Thus, the primary objective of this review is to boost the development of a gas-phase system for practical application. Although some previous documents have summarized the important factors on only the configuration of gas-phase systems [33–36], we believe it is important to recall lessons in order to optimize catalysts for gas-phase reactors from previously studied catalysts in liquid-phase reactions, because tremendous advances have been achieved. Then, this review manuscript will summarize the research on the gas-phase reactors. Particularly, we will briefly introduce the representative type of gas-phase reactors and provide general reviews by grouping previous reports according to the final products such as formate, carbon monoxide, and hydrocarbon. Through this work, we finally outline the significant issues and perspectives of the gas-phase reactor systems toward a practical application level.

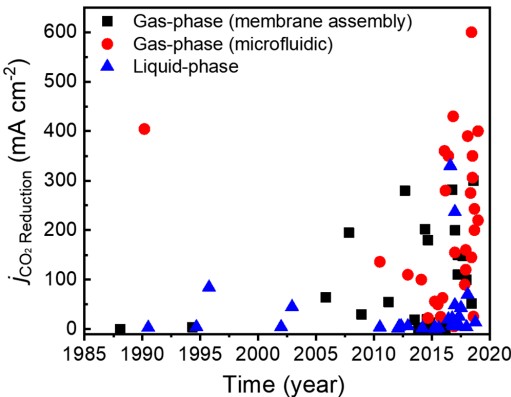

**Figure 4.** The trend of partial current density for $CO_2$ reduction ($j_{CO2\ Reduction}$) in the case of liquid- [3–8,10,11,13,16,37–53] and gas-phase reactors [15–18,23–32,54–93]. The gas-phase reactors are divided into membrane assembly and microfluidic types.

## 2. CO$_2$ Reduction Reaction with Liquid-Phase Reactor

Numerous research groups have carried out $CO_2$RR studies in aqueous solutions by using their own home-made liquid-phase reactors [3,6,7,43,45–48,94,95]. Figure 5 shows the liquid-phase reactor typically called a H-type cell, which consists of two chambers for the cathode and anode, separated by an ion exchange membrane. $CO_2$ gas is supplied to the electrolyte via a porous glass frit placed at the cathode side for improving the solubility of the $CO_2$ gas in the aqueous solution [96]. This type of liquid-phase reactor has been widely used as a tool for evaluating $CO_2$RR catalysts, because of its versatile configuration for various types of catalyst electrodes and its easy separation of products. However, the polarization losses caused by the low solubility of $CO_2$ in the electrolyte, and the mass transfer limitation of $OH^-$ near the electrodes, inevitably limit the productivity of $CO_2$RR. Despite this drawback for practical application, the liquid-phase reactor has served as an important platform to reveal the various factors for catalytic activity, thereby contributing to material improvements for $CO_2$RR. Herein, we provide a brief review of the major factors (structuring, surface tailoring, and electrolysis environments) contributing to a higher catalytic activity, as depicted in Figure 6 [4,8,13,16,42,44,97,98]. For example, we describe the influence of catalytic properties by fabricating nano- or micro-structures of catalysts, tailoring the surface with the immobilization of chemical species, and the pH change of electrolytes. This work will provide an important guideline for designing enhanced catalytic materials in gas-phase systems.

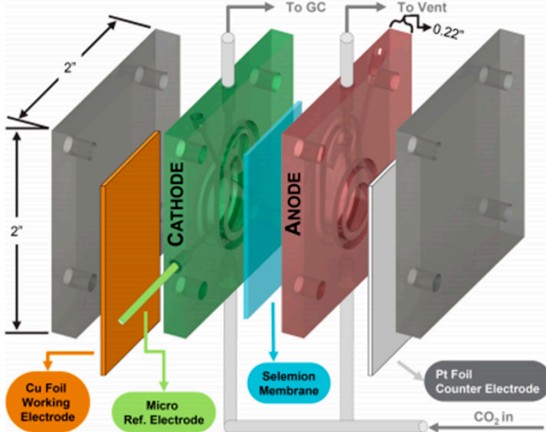

**Figure 5.** Schematic of a typical home-made electrochemical reactor for liquid-phase $CO_2$ reduction. A liquid-phase reactor consists of two chambers for the anode and cathode, a membrane, and so on. The volume and area depend on the purpose of the user. Reprinted with permission from the authors of [96].

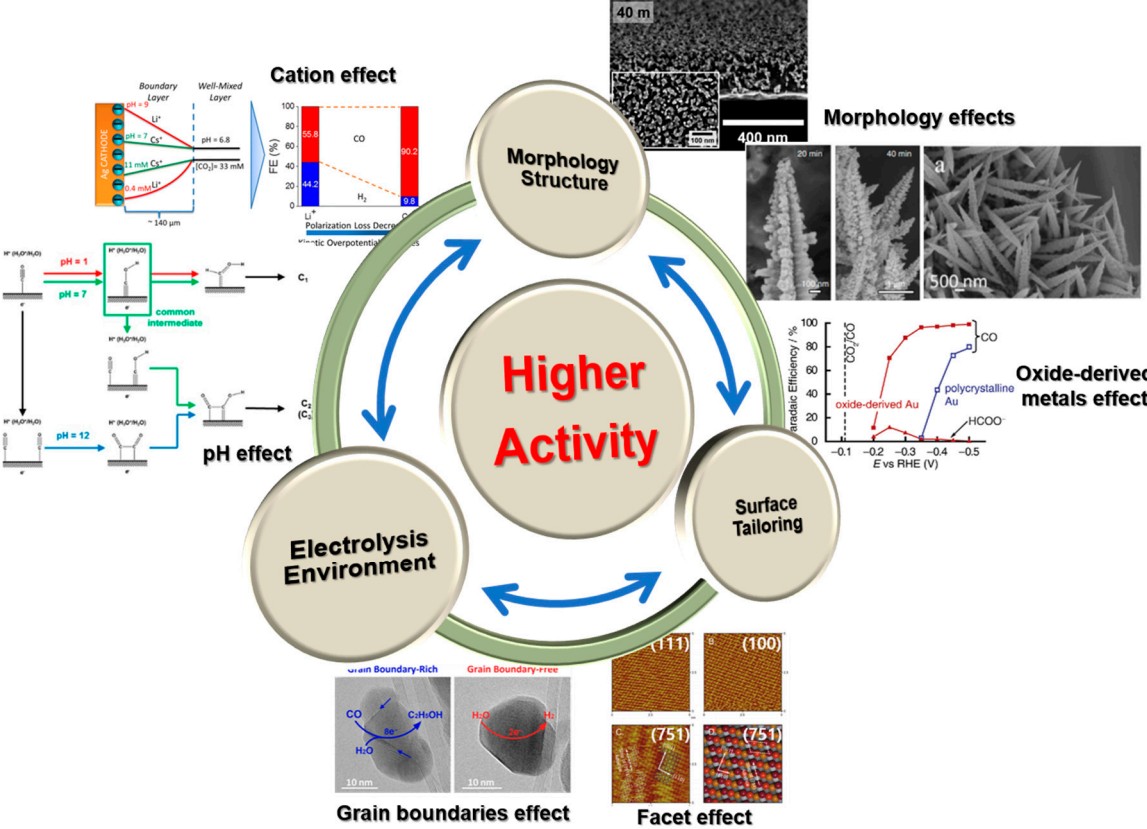

**Figure 6.** Major factors to improve the overall catalytic activities, with representative research examples. Reprinted with permission from the authors of [4,8,13,16,19,42,44,98]. Copyright (2012) and (2016), American Chemical Society for [4,19,42,98].

## 2.1. Nano- and Micro-Structures

The most intensively studied approach is nanostructuring technology, to enhance the activity for not only C1, but also higher-order carbon products, using Au-, Ag-, and Cu-based materials. Sargent and co-workers fabricated an Au nanoneedle structure that can induce a high local electric field around the end of the nanoneedle tip, eventually leading to a high $CO_2$RR performance [13,16]. That is, the Au nanoneedle structure enabled an increase in the available reagent concentration for a higher current density and lowered overpotential for CO production (22 mA cm$^{-2}$ at $-0.35$ V (vs. reversible hydrogen electrode (RHE)) [13]. In addition, they demonstrated a similar concept with a Cu catalyst electrode synthesized via electro-redeposition (ERD), forming a sharp morphology and a Cu oxidation state ($Cu^+$) in order to overcome a low local concentration of $CO_2$, and slow kinetics in the multi electron/proton transfer reaction [13]. As a result, the ERD Cu exhibited an extremely high partial current density for $C_2H_4$ of 22 mA cm$^{-2}$ at $-1.0$ V (vs. RHE).

Nanostructuring methods to increase the intrinsic active sites of catalytic materials also have been reported [4–6,8,12,13,16]. The Kanan research group synthesized oxide-derived (OD) metal catalysts with Au and Cu (OD-Au and -Cu) by electrochemically reducing Au and Cu oxide [4,5]. OD-Au showed a lower overpotential property and higher Faradaic efficiency for CO evolution than bare polycrystalline Au foil. OD-Cu formed by the reduction of annealed $Cu_2O$ also produced formic acid, CO, and $C_2$ hydrocarbons, with a higher activity than normal Cu. It was suggested that oxide-derived metals possess abundant grain boundaries that can stabilize the intermediates of $CO_2$ reduction. Meanwhile, our research group developed nanostructured Au on a 200-nm-thick film structure by mild electrochemical processes. It exhibited a 96% CO Faradaic efficiency at 480 mV overpotential, while the maximum selectivity of a bare Au thin film was about 70% [8]. Increased grain boundaries on the nanoporous Au catalysts were also found to be active sites for $CO_2$RR.

Recently, the Cu catalysts fabricated by oxidation/reduction cycling in the presence of chloride anions dramatically enhanced the selectivity of higher-order carbon products ($C_2H_4$ and $C_2H_5OH$) [95]. This property is also attributed to the production of a number of grain boundaries through a reduction process of $Cu_2O$. On the other hand, the Zhang group utilized a highly porous carbon substrate for a hierarchical mesoporous $SnO_2$ nanosheet. As a result, a superior partial current density for formate of 45 mA cm$^{-2}$, with a 89% Faradaic efficiency and stable operation of 24 hours, was achieved. This outstanding performance is attributed to the large surface area and facile charge and mass transfer by a hierarchical structure [99].

## 2.2. Surface Tailoring

Many research groups have attempted to control the surface orientation, because the facet sensitivity for $CO_2RR$ has been widely studied in Cu catalysts [9,44,100,101]. Recently, Hahn et al. engineered the surface of Cu catalysts by using epitaxial growth in the (100), (111), and (751) orientations, with the physical vapor deposition method on a large electrode (~6 cm$^2$) [44]. To probe the correlation of the surface index and the electrochemical catalytic activity, in-situ scanning tunneling microscopy was conducted in this study. It was confirmed that undercoordinated sites lead to higher C–C coupling. That is, the Cu (100) and (751) electrodes are more active for $C_{2+}$ products than Cu (111) films. In particular, a Cu (751) thin film that includes a heterogeneous kinked surface with (110) terraces activated the reaction of $>2e^-$ oxygenate production at a low overpotential region. On the contrary, the catalytic activity has been enhanced by directly modifying the surface with various chemical additives [11,20,21]. Our group reported an electrodeposited Au film functionalized with CN or Cl species, and its performance was highly improved compared with a bare Au film in terms of selectivity and overpotential [11]. We suggest that the enhanced $CO_2RR$ performance originated from the stabilized intermediates of $CO_2RR$ by van der Waals interactions between the intermediates, and attached anionic species via computational investigations. Kim et al. also synthesized surface-modified Ag nanoparticles by using anchoring agents containing cysteamine or amine, respectively, achieving a highly selective CO formation [20,21]. Via DFT calculations, both anchoring groups were revealed to assist the $CO_2RR$ at the catalyst surface, by stabilizing intermediates and destabilizing hydrogen binding, and suppressing the hydrogen evolution reaction (HER). The molecular catalysts of porphyrin also significantly improve the $CO_2RR$ activity with Ru and Co catalysts [102,103]. It influences the thermodynamic stability of key intermediates of $CO_2RR$, thereby increasing the catalytic activity.

## 2.3. Electrolysis Environments

Changes in the electrolysis environment, and the tuning of catalytic materials, also have a great effect on the $CO_2RR$. The pH of the electrolyte is one of the most critical parameters in the reaction mechanism of $CO_2RR$. Schouten et al. investigated the pH dependent reaction pathways to study the $CO_2RR$ mechanism in Cu (111) and Cu (100) single crystal electrodes [97]. They proposed the following reaction pathways for hydrocarbon with respect to pH and the Cu surface: (1) a pH-dependent mechanism for $CH_4$ formation involving CHO intermediates on Cu (111) and (2) a pH-independent mechanism for $C_2H_4$ formation via CO dimerization on Cu (100). This was consistently observed in the pioneering results of the Hori research group.

Moreover, a computational mechanistic analysis of the pH dependence for the CO reduction, which is the key intermediate species of $CO_2RR$ for $C_2$ production, was carried out in order to fundamentally understand the complicated reaction pathways in different pH conditions [19]. As a result, the dominantly involved intermediates vary depending on the pH value. In an acidic condition, the $CH_4$ product proceeds via COH and COHO intermediates. However, at the neutral region, the pathway for $C_{2+}$ products can be activated by sharing common COH intermediates on the surface, whereas the CO dimerization at a higher pH leads to long chain carbon products.

The cation size in the electrolyte can alter the $CO_2RR$ activity. As the size of the electrolyte cations becomes larger, the $CO_2RR$ rate was shown to increase compared with the case of the smaller cations on both Cu and Ag catalysts [42]. This phenomenon might occur because the lowered pH near the surface with larger cations promotes increasing the available $CO_2$ concentration nearby the metal electrodes. Finally, the utilization of an ionic liquid is widely known as a facile method to increase the $CO_2$ solubility and catalytic activity at the same time [39,40,104,105]. Salehi-Khojin and co-workers recently studied a transition metal dichalcogenide (TMD) nanostructure for $CO_2RR$, with an ionic liquid [39,40,105]. Molybdenum disulphide and tungsten diselenide nanoflake showed high CO current densities of ~4.5 mA cm$^{-2}$ at a remarkably low overpotential of 54 mV in a diluted solution of 1-ethyl-3-methylimidazolium tetrafluoroborate (EMIM-BF$_4$) ionic liquid [39]. Interestingly, the nanostructured TMD catalysts exhibited a significantly higher catalytic activity than the Ag nanoparticles and bulk Ag, known as efficient $CO_2$-to-CO electrocatalysts, indicating the remarkable capability of the TMD catalyst for $CO_2RR$. This study revealed a synergetic effect between the EMIM-BF$_4$ ionic liquid and TMD electrocatalysts, by facilitating the transport of $CO_2$ to the intrinsic active sites of TMD catalysts through a complex reaction.

## *2.4. Synergistic Effects*

Nano-structuring materials can affect not only the intrinsic catalytic property with increased active sites, but also the electrolysis environment near the catalytic surface. For example, Yogesh and co-workers fabricated highly ordered, uniform Au- and Ag-inverse opal (Au- and Ag-IO) structures with a controlled thickness in order to analyze the correlation between the structured catalysts and the local environment [106,107]. Au- and Ag-IO can systematically tune the selectivity of $CO_2RR$, reaching a 99% CO selectivity at optimal thickness. They demonstrated that the enhanced selectivity of CO resulted from the suppressed HER by a high local pH inside the Au- and Ag-IO structures, due to the diffusional gradients of $OH^-$ rather than the increased surface area. Furthermore, we evaluated the catalytic activity for $CO_2RR$ by using controllable Cu meso-structures, which produce various hydrocarbons and alcohols via a multi electron and proton transfer reaction [10]. It showed highly reduced overpotentials of $C_2$ products ($C_2H_4$ and $C_2H_5OH$) with layers of optimal thickness, due to a highly formed local pH near the catalyst surface. We suggested that the electrolysis environment changes by Cu meso-structures can activate specific target reaction pathways and experimentally proposed reaction pathways via observation of the $C_2H_2$ intermediate. Likewise, Ma et al. showed that denser and longer Cu nanowires (NW) can activate $C_2H_4$ formation at a fixed potential by preferred CO dimerization, as a result of a high local pH [108]. Together with the high local pH in structured catalysts, increasing the retention time of intermediates can also influence the formation of $C_2$ products. Dutta et al. demonstrated highly selective $C_2$ product ($C_2H_4$ and $C_2H_6$) formation, reaching a 55% Faradaic efficiency by fabricating mesoporous Cu foam [38]. The temporal trapping of CO and $C_2H_4$ intermediates in the mesoporous catalysts can promote a further reduction, and thereby induces more $C_2H_4$ and $C_2H_6$ production. Similarly, Yang et al. proved that the controlled width and depth of the nano-scale morphology can affect the selectivity of $C_2H_4$ and $C_2H_6$, because of the high local pH and the increased retention time of the diffused intermediates [41].

Recently, Lum et al. further optimized the performance of OD-Cu electrodes for more C–C coupling by preparing various types of OD-Cu with the use of an electrolyte containing $Cs^+$, which was reported to increase the $CO_2$ concentration near the surface. They finally achieved an exceptionally high $C_{2+}$ Faradaic efficiency of 70% [109]. The optimized OD-Cu also has an appropriate surface roughness, inducing a high local pH at the surface for facile $CO_2RR$. In addition, our research group systematically studied a controllable Au nanostructure with various electrochemical treatment conditions, and evaluated the $CO_2RR$ properties [12]. We found the following two significant factors contributing to a higher $CO_2RR$ rate: (1) increased grain boundaries inside Au nanoparticles for a lower overpotential of $CO_2RR$, as previously reported by many groups, and (2) the structure shape for facile diffusion of the reactant and evolved products for high local pH.

## 2.5. Summary of Liquid-Phase CO$_2$ Reaction

With the systematical control of the morphology, surface, and electrolysis environments, numerous studies have been conducted to elucidate the electrochemical CO$_2$ reduction reaction. Based on an analysis of these intrinsic and extrinsic properties, the CO$_2$RR activities and stability of the tuned electrocatalysts have been significantly improved in the liquid-phase reactor. Table 1 presents a summary of representative research mentioned in the above sections. Despite efforts for enhancing the CO$_2$RR performance and fundamental understanding with a liquid-phase reactant, the achieved performance to date is still below the practical level, mostly because of the CO$_2$ solubility limitation. Thus, we outline previous studies with gas-phase reactors and seek opportunities for the further development of electrochemical CO$_2$ reduction technology in the next chapter.

**Table 1.** Representative catalysts and their CO$_2$ reduction reaction (CO$_2$RR) activity. EMIM-BF$_4$—1-ethyl-3-methylimidazolium tetrafluoroborate.

| Catalyst | Target | Note | Ref |
|---|---|---|---|
| Au nanoneedle | CO | Nano- and micro-structure, field-induced reagent concentration 22 mA cm$^{-2}$ at −0.35 V (vs. RHE) | [13] |
| Pd nanoneedle | Formate | Nano- and micro-structure, field-induced reagent concentration 10 mA cm$^{-2}$ at −0.2 V (vs. RHE) | [13] |
| Electro-redeposited Cu | C$_2$H$_4$ | Nano- and micro-structure, sharp-tip morphology, field-induced reaction concentration 22 mA cm$^{-2}$ at −1.0 V (vs. RHE) | [16] |
| Oxide-derived Au | CO | Nano- and micro-structure, increased grain boundaries (active sites) >96% Faradaic efficiency for CO at −0.35 V (vs. RHE) | [4] |
| Nanoporous Au | CO | Nano- and micro-structure, increased grain boundary (active sites) >96% Faradaic efficiency for CO at −0.59 V (vs. RHE) | [8] |
| SnO$_2$ nanosheet | Formate | Nano- and micro-structure, facile charge and mass transfer 45 mA cm$^{-2}$ at −0.88 V (vs. RHE) | [12] |
| Cu (100), (111), and (751) thin films | >2e$^-$ oxygenates | Surface tailoring–facet control for C–C coupling Cu (751), (100) have higher oxygenate/hydrocarbon ratios than that of Cu (111) | [44] |
| Functionalized Au | CO | Surface tailoring—surface functionalization with anion, stabilization of intermediates >92% Faradaic efficiency for CO at −0.39 V (vs. RHE) | [11] |
| Ag foil | CO | Electrolysis environments—effect of cation size of electrolytes 90.2% Faradaic efficiency for CO at −1.0 V (vs. RHE) in 0.1 M CsHCO$_3$ | [42] |
| WSe$_2$ nanoflake | CO | Electrolysis environments—ionic liquid (4% EMIM-BF$_4$ electrolyte) increased available reactants 320 mA cm$^{-2}$ at −0.764 V (vs. RHE) | [39] |
| Au inverse opal structure | CO | Synergetic effects—nanostructure and high local pH 99% Faradaic efficiency for CO at −0.51 V (vs. RHE) | [106] |
| Oxide-derived Cu foam | C$_2$H$_4$ + C$_2$H$_6$ (C$_2$) | Synergetic effects—nanostructure, oxide-derived Cu, high local pH 55% Faradaic efficiency for C2 at −1.0 V (vs. RHE) | [38] |
| Oxide-derived Cu | C$_{2+}$ | Synergetic effects—increased active sites and CsHCO$_3$ electrolytes for the optimization of C–C coupling 70% Faradaic efficiency for C$_{2+}$ at −1.0 V (vs. RHE) in CsHCO$_3$ electrolytes | [109] |

### 3. CO$_2$ Reduction Reaction with Gas-Phase Reactor

*3.1. Type of Gas-Phase Reactor Cell*

As described in the introduction section, the utilization of dissolved $CO_2$ in an aqueous solution as reactants for $CO_2$RR brings mass transfer limits that hinder scale-up. For the continuous supply of the $CO_2$ reactants, there are two representative approaches with different reactor architectures, as depicted in Figure 7. For gas-phase reactors, a gas diffusion electrode (GDE) composing the catalyst surface and highly porous substrate layers is an essential component for a cathodic catalyst electrode and the pathway of the gaseous $CO_2$ reactant to the catalyst surface.

The most widely studied design, a membrane electrode assembly (MEA) reactor developed from fuel cell systems, includes a polymer electrolyte membrane (PEM) for ion exchange between the cathode and anode. The typical configuration of the MEA reactor is shown in Figure 7a. In fact, some transformed designs have been also presented by various research groups, and the detailed structures are well specified in a previously reported review paper [33]. Mainly, various types of PEMs, such as an anion, cation exchange membrane, and bipolar membrane, have been adopted, and the PEM is the key component to determine the overall performance of the $CO_2$ electrolysis cell [24,59,92,93]. Also, in the MEA system, the suppression of side products causing the deterioration of the membrane and the phase of the delivered $CO_2$ are significant issues to improve the $CO_2$RR activity [26,33]. On the other hand, a microfluidic reactor type was first proposed by Cook et al., with the configuration of a gas diffusion region/Cu catalysts/aqueous electrolyte [60]. In this cell configuration, the PEM is not a necessary component, and was not included in most previous demonstrations, as shown in Figure 7b. The cathode and anode are separated by an electrolyte stream in this case. The membrane is optionally equipped to prevent the crossover of evolved liquid products to the anode side. Although the *iR* loss that originated from the membrane degradation during electrolysis is not a concern, the invasion of the electrolyte into the porous layer of the GDE should be prevented for stable operation. This setup also enables the insertion of a reference electrode for a three-electrode configuration.

Both of the gas-phase reactor devices have recently shown feasibility for a high current density for target products, but further improvement is still required, by addressing various issues for industrial applications. It is imperative to address the catalytic materials determining the overpotential and target product selectivity, as well as the above-mentioned issues from the reactor architectures. In the following sub-sections, we derive the perspectives by reviewing the previous reports for three different final products in MEA and microfluidic gas-phase $CO_2$RR reactor types.

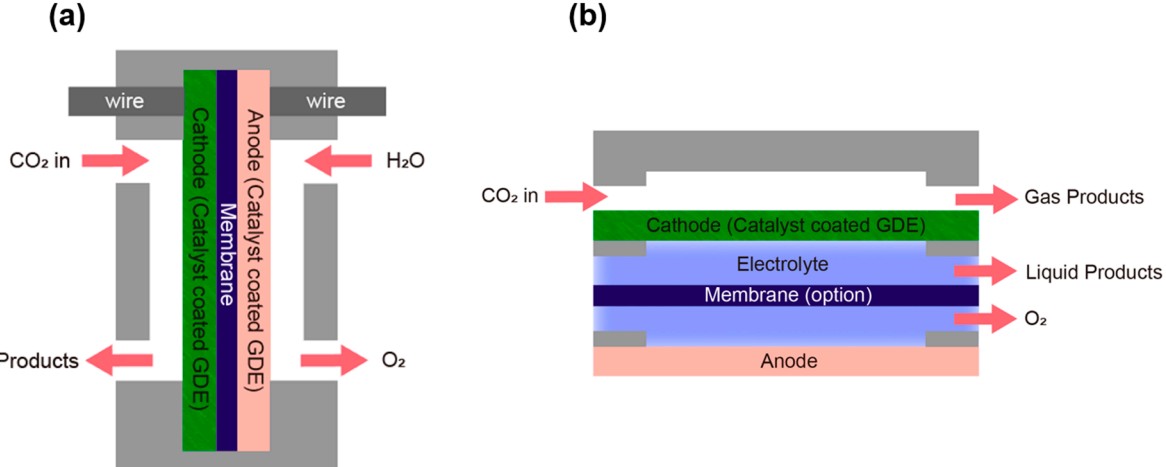

**Figure 7.** The cell configuration comparison for gas-phase $CO_2$ reduction reaction: (**a**) membrane electrode assembly (MEA) and (**b**) microfluidic type.

## 3.2. Membrane Electrode Assembly (MEA) Reactor

### 3.2.1. Formate

Formate production has been intensively studied with MEA reactors, in particular with Sn electrocatalysts [24,26,28,58,64,67,86,88–91]. Nevertheless, the current density achieved in most studies is not adequate for scalable technology. Lee et al. recently presented a high partial current density for formate production ($j_{HCOOH}$) of 51.9 mA cm$^{-2}$ with a 93.3% Faradaic efficiency [26]. They fabricated a MEA flow cell without a catholyte by using vaporized $CO_2$ gas as a reactant, as shown in Figure 8a. The catholyte-free $CO_2$RR with a commercial Sn catalyst GDE shows higher $CO_2$RR activities, such as productivity and durability, compared with using a catholyte of 1.0 M KCl (Figure 8b). Meanwhile, the utilization of a buffer layer between the proton exchange membrane and the cathode can also be an effective strategy to increase the current efficiency for HCOOH, as depicted in Figure 8c [58]. The authors suggested that the buffer layer plays a key role in providing a sufficient potential to the surface of the catalysts for $CO_2$ reduction, because a quite large potential drop occurs via the membrane in conventional cell structures (Figure 8d). As a result, the Sn GDE fabricated with commercial particles exhibited roughly 150 mA cm$^{-2}$ and a 60% selectivity for HCOOH, whereas the selectivity is only about 5% without the buffer layer. In addition, the optimization of the Sn nanoparticles on the GDE was also explored in order to attain a higher activity by carbon-supported Sn nanoparticles [86]. Because of the carbon components, the catalytic layer with Sn nanoparticles on a gas diffusion substrate layer was formed with a higher thickness and porosity, as compared to only a pure Sn catalytic layer. Accordingly, it achieved a substantially higher $j_{HCOOH}$ of 110 mA cm$^{-2}$ with a 70% Faradaic efficiency. Although these works have demonstrated the feasibility of MEA reactors toward $CO_2$RR, formate production with MEA reactors gives rise to an important concern regarding stability. Wang et al. reported that the accumulation of the produced formic acid can attack the membrane performance during long-term $CO_2$ electrolysis [91]. These stability issues critically impede commercial use, although a large scale MEA cell was previously demonstrated [64]. The Oloman group increased the Sn-based GDE with a geometric area of 320 cm$^2$, and attained a high $j_{HCOOH}$ of 195 mA cm$^{-2}$. However, the operating conditions of $CO_2$ electrolysis by a large size cell is 5.9 atm and 314 K, causing additional costs. It also showed a gradual decrease of the catalytic activity over only 100 min.

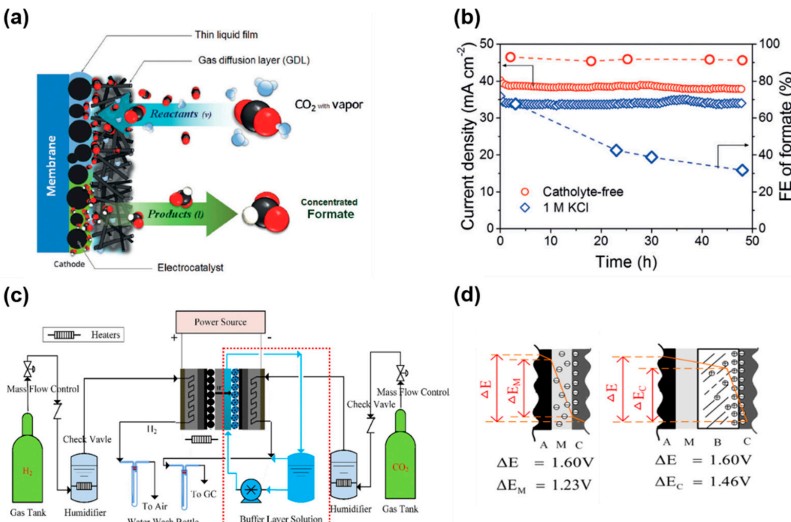

**Figure 8.** MEA reactor configuration and effects on a gas diffusion electrodes (GDE) incorporating Sn catalysts. (**a**) GDE system operated in the condition of catholyte-free $CO_2$ reduction, and (**b**) comparison of the catalytic performance under catholyte-free and 1.0 M KCl catholyte conditions. (**c**) Schematic of the MEA setup with buffer layer, and (**d**) a diagram of the potential distribution with and without the buffer layer (A—anode; M—membrane; C—cathode; B—buffer layer). Reprinted with permission from the authors of [26,58]. Copyright (2017) American Chemical Society for [58].

### 3.2.2. Carbon Monoxide

The production of carbon monoxide (CO) with a MEA configuration by using mostly a Ag catalytic material has received considerable attention. Comparing the productivity (i.e., partial current density) with the formate production, most of the previous studies have shown a higher partial current density for CO ($j_{CO}$) over mA cm$^{-2}$. The performance of MEA systems with Ag-GDE was also dependent on various operating parameters, such as pH, temperature, and pressures. Figure 9 describes the different $CO_2$ reduction reaction process on Ag GDE catalysts without and with a pH buffer layer of $KHCO_3$ between the cathode and Nafion membrane [55]. The incorporation of a pH buffer layer in MEA systems dramatically enhanced the CO selectivity to ~80%, effectively suppressing the $H_2$ evolution, while almost no CO was generated without the buffer layer. In the case of using the pH buffer layer, the bicarbonate in the buffer layer is expected to couple with the proton forming $CO_2$ and $H_2O$. This leads to an increased local pH near the cathode, whereas the protons from the anode can be directly moved to the cathode side without a buffer layer. As a result, a more favorable pH condition for electrochemical $CO_2$ reduction near the surface of the catalysts is attributed to a high $CO_2$RR activity. Nevertheless, the maximum $j_{CO}$ was not impressive, being only 30 mA cm$^{-2}$, with a decrease of the CO Faradaic efficiency by 3% per hour. Meanwhile, Dufek et al. observed a five times higher produced CO quantity ($j_{CO}$ of ~280 mA cm$^{-2}$ and CO selectivity of 92%) using an Ag-based GDE by elevating the pressure (24.7 atm) relative to the ambient pressure [65]. The benefits of increasing the temperature up to 90 °C were also assessed in an operating cell. At 90 °C, the cell voltage dropped from ~3.7 V to below 3.0 V with an applied current density of 225 mA cm$^{-2}$, achieving a 50% energy efficiency. Although these works have shown that some parameters should be significantly controlled in order to improve the performance of MEA systems converting $CO_2$, research on the ion exchange membrane reactor, which is a key component in the MEA, is lacking. Indeed, it was reported that optimization of the membrane composition is necessary for a more efficient $CO_2$ reduction to CO [59,110,111]. However, the current density was quite low (below 10 mA cm$^{-2}$) to fulfill the economic viability.

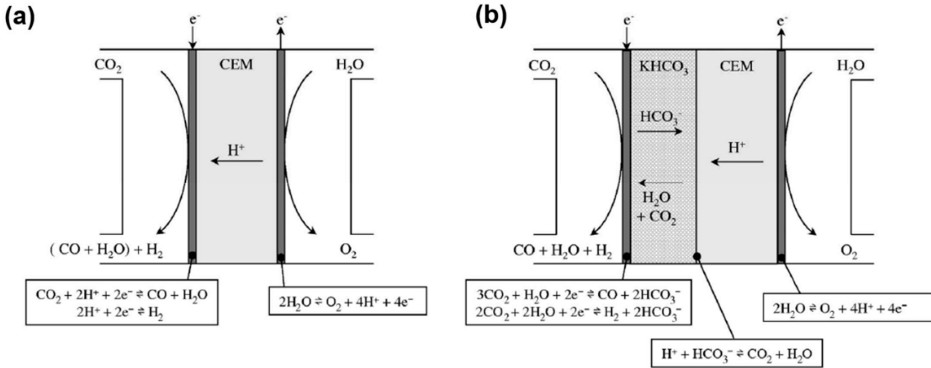

**Figure 9.** The schematics for MEA cells adopting a cation-exchange membrane (**a**) without and (**b**) with the $KHCO_3$ buffer layer. Reprinted with permission from the authors of [55].

MEA reactors configured with a bipolar membrane have recently been reported as showing a highly improved activity [57,112]. For example, the Berlinguette research group achieved a higher current density of 200 mA cm$^{-2}$ with the silver-based catalyst and the bipolar membrane. The bipolar membrane drives both $H^+$ and $OH^-$ ions toward the anode and cathode, respectively, enabling a constant pH for higher stability. A bipolar membrane design was highly required when the co-electrolysis cell configuration is operated with an alkaline and an acidic condition at the cathode and anode, respectively. It can prevent the parasitic $CO_2$ transport via the membrane to the anode side, maintaining a high $CO_2$RR selectivity [112]. For the improved properties of the anion exchange membrane, the company Dioxide Materials has developed new membrane materials incorporating an imidazolium group, which showed a superior effect to decrease the overpotential for $CO_2$ to CO, and completely suppress $H_2$ evolution [24,25,61,87,104]. Because of the high cost of using ionic

liquid for imidazolium, they invented an alternative membrane by using imidazolium-functionalized stylene and vinylbenzyl chrolide-based polymers, called Sustainion. The structure of the developed membrane is depicted in Figure 10a. Figure 10b shows the polarization curve of the MEA systems with Sustainion (PSTMIM-Cl) and commercial AMI-7001. The Sustainion membrane has a much higher current density than AMI-7001 under the supply of $CO_2$ into the cathode part. Surprisingly, it showed excellent durability, maintaining over a 90% CO selectivity for about six months at 50 mA cm$^{-2}$, as exhibited in Figure 10c. In addition, the Sustainion membrane was confirmed to be very stable in an alkaline solution of 1.0 M KOH, whereas most commercial membranes are unstable under the same conditions. The company recently presented a $CO_2$ electrolysis cell operating at 200–600 mA cm$^{-2}$, a at cell voltage of 3.0–3.2 V with a 95%–99% CO selectivity [87]. These works show the significance of the further work on PEM, which is a key in the MEA reactor to contributing to the commercialization of the $CO_2$ reduction technology.

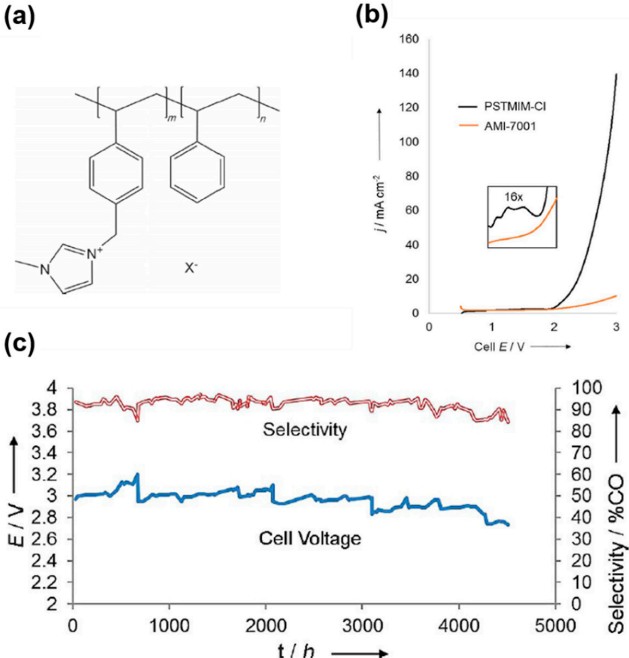

**Figure 10.** The general characteristics of Sustainion PSTMIM membrane (**a**) chemical structure of the PSMIM membrane. (**b**) Comparison of current density for PSMIM and AMI-7001 membranes. (**c**) The results of a long-term $CO_2$ reduction test with the PSMIM membrane. Reprinted with permission from the authors of [24].

### 3.2.3. Hydrocarbons

As we discussed, with regard to electrochemical $CO_2$ reduction based on a liquid phase, in Section 2, only Cu has a capability to allow for the evolution of hydrocarbon products. In fact, there have been few previous examples with Cu-GDE in MEA-based $CO_2$ electrolysis cells, likely as a result of the difficulty of the selective production of target chemicals [54,59,82,92,93,113–115]. Most works could not show the advantages of gas-phase $CO_2$ reduction with poor productivity. Recently, de Lucas-Consuegra and co-workers demonstrated a Cu-based GDE with the use of a proton exchange membrane (Sterion) [54]. Although the current density for $CO_2$RR is far from the economically required level, it is interesting that highly value-added products such as methanol and acetaldehyde were evolvable as the products of $CO_2$ electrolysis. They investigated the dependence of carbon supports on the electrocatalytic activity of the MEA system by preparing various Cu-based GDE samples with graphite, activated carbon, and carbon nanofibers. Among them, the activated carbon support enabled the largest productivity in all of the explored conditions, as a result of offering the highest surface area. Interestingly, methanol was the main product with the graphite-based carbon

support, whereas acetaldehyde was dominantly observed as the final product for the other supports. Although the reasons are not revealed here, we believe that understanding the reaction mechanism is necessary for further improvement, given the lessons from the electrochemical $CO_2$ reduction in a liquid-phase reactor. In the early stage of the $CO_2$ reduction research, Komatsu et al. demonstrated a gas-phase electrochemical $CO_2$ reduction by preparing the Cu-solid polymer electrolyte composite electrodes [93]. They showed a low partial current density for a $CO_2$RR of 3.96 mA cm$^{-2}$, but the selectivity of the $C_2H_4$ product was found to increase with a Nafion-based electrode compared to Selemion, because of the higher proton supply by Nafion. A similar observation was reported by Hori et al. for the electroreduction of $CO_2$ by using Cu electrodes in a $KHCO_3$ aqueous solution.

Moreover, Merino-Garcia et al. analyzed the potential-dependent distribution of hydrocarbons using Cu-nanoparticles supported on porous carbon papers with a modified MEA configuration [114]. They focused on the influence of the particle size of Cu for productivity and selectivity. As a result, the smaller Cu nanoparticels (25 nm) showed a remarkably high Faradaic efficiency of ethylene (92.8%). In particular, they used a reference electrode for controlling the potentials in the MEA system in the anode compartment, by incorporating the anolyte between anode and membrane. So, they evaluated the effect of the catalyst loading and Cu nanoparticle size for $CH_4$ and $C_2H_4$ formation. It indicated the significance of the control of the potential to precisely evaluate the kinetics of $CO_2$RR at the catalyst surface in a MEA system. On the other hand, the Irvine research group demonstrated the co-electrolysis of $CO_2/H_2O$ by using an oxygen-ion or proton conducting solid oxide electrolyzer [116,117]. In this configuration, the electrolyzer was operated at a high temperature of 614 °C using a solid electrolyte. Despite the low Faradic yield for $CH_4$, they achieved a very high rate production of 1.5 A cm$^{-2}$.

### 3.3. Microfluidic Reactor

#### 3.3.1. Formate

Research on microfluidic reactors has been extensively carried out by the Kenis group with various approaches [27,30–32,68–75,78,79,118]. Among them, formate production has been targeted by using mainly Sn-based GDE catalysts, which is in common with the MEA reactor type. Figure 11a shows the microfluidic reactor design constructed by Kenis and co-workers; it contains a flowing electrolyte stream between the anode and cathode, without the ion exchange membrane constructed by Kenis and co-workers [27]. The electrolyte stream can provide the flexibility of the pH and electrolyte type for favored environments. It also has the advantage of the facile separation of liquid products from inside of the reactor. They varied the pH value for the best condition with Sn-GDE in a range of 4–10, by adjusting the electrolyte, as shown in Figure 11b. The acidic condition shows the highest $CO_2$RR activity with a $j_{HCOOH}$ of 136 mA cm$^{-2}$ at a cell potential of 3.8 V. It was noted that a lower pH was avoided because Sn is dissolved in a highly acidic environment. Meanwhile, the effects of the loading amount and size for the Sn catalyst were reported by Castillo and co-workers. They tried to find the optimal conditions for Sn particle loading on a carbon support. The best performance (Faradaic efficiency of 70% at 90 mA cm$^{-2}$) was observed when the smallest particle size (150 nm) and loading amount (0.1–0.75 mg cm$^{-2}$) in the tested range were applied on carbon paper. Notably, the larger loading caused particle agglomeration, thus emphasizing the importance of well-dispersed particles for a higher activity. Recently, Liang et al. synthesized much smaller sized (<5 nm) $SnO_2$ nanoparticles via a cryo-exfoliation method, achieving a high crystallinity [118]. The fabricated $SnO_2$ nanoparticles exhibited a two to three times larger current density as compared to the $SnS_2$ sheets and $SnO_2$ bulk samples. They achieved a partial current density of a $CO_2$ reduction of 145 mA cm$^{-2}$ at −1.21 V (vs. RHE). Interestingly, $CH_4$ and $C_2H_4OH$ were additionally shown as products with a ~10% Faradaic efficiency. This is attributed to the unique structures of the interconnected $SnO_2$ nanoparticles possessing many grain boundaries between the nanoparticles, which may act as active sites for hydrocarbons and oxygenates.

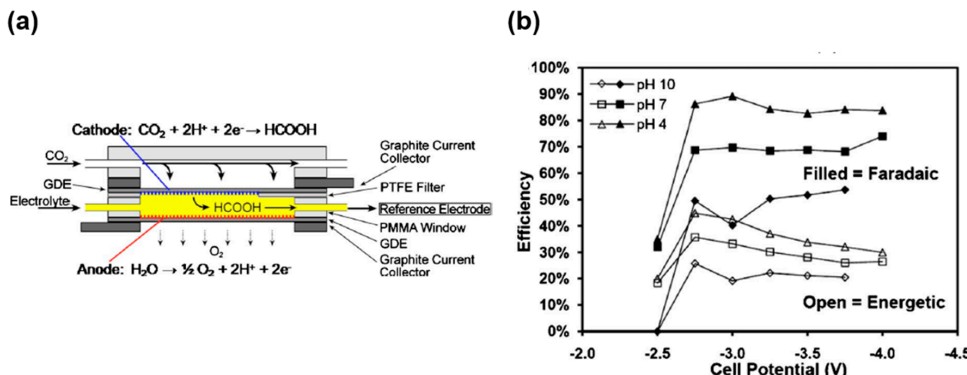

**Figure 11.** (**a**) Microfluidic type reactor cell. (**b**) Plot of Faradaic efficiency of formate production with Sn catalysts with regard to pH condition. Reprinted with permission from the authors of [27].

In addition to the Sn catalysts, 2D metal catalysts of bismuth oxyhalides were recently reported for formate production by the Sargent research group [15]. The catalyst samples were prepared by coating BiOBr onto carbon paper, followed by an annealing process in an inert atmosphere. The BiOBr-templated catalysts were then obtained after the electroreduction process in $CO_2$-saturated $KHCO_3$ solutions, so as to remove the precursor (Figure 12a). Prior to the GDE application of the BiOBr catalysts, the $CO_2$RR activity was evaluated in a normal H-type electrochemical cell for the liquid-phase reaction. Even under the influence of a mass-transport limitation, they achieved a high current density of ~80 mA cm$^{-2}$ with almost a unity Faradaic efficiency for the formate production, and a remarkably stable operation for 65 hours, as shown in Figure 12b. These features allowed for much higher current densities of 200 mA cm$^{-2}$ in a 2.0 M $KHCO_3$ electrolyte in a microfluidic cell configuration. A direct comparison between the liquid- and gas-phase thus clearly shows the advantages and importance of gas-phase reactors for high productivity.

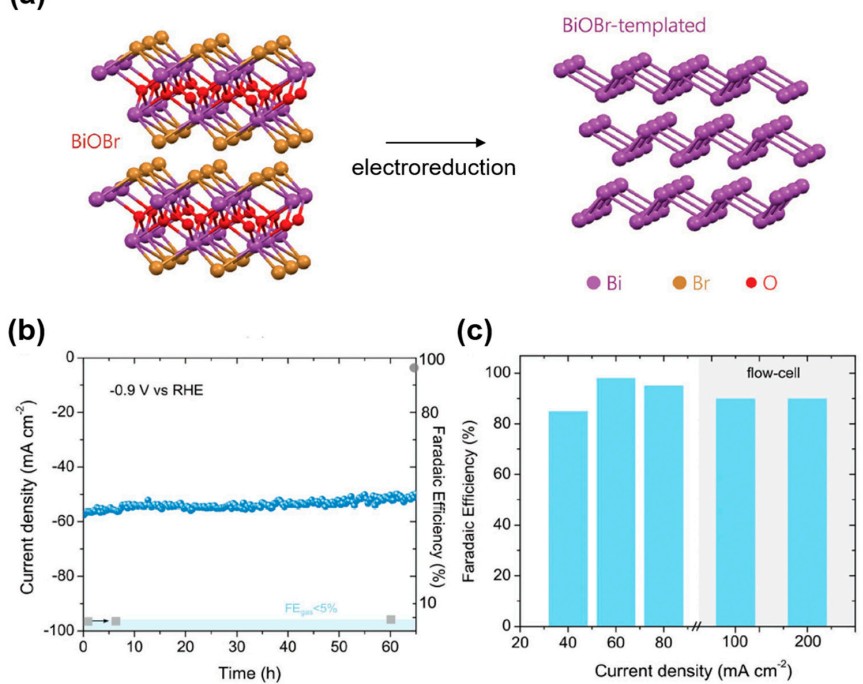

**Figure 12.** The characteristics of bismuth oxyhalide-templated catalysts. (**a**) Atomic structures of BiOBr and BiOBr-templated via the electroreduction process. (**b**) Long-term operating examination of liquid-phase $CO_2$ reduction reaction. (**c**) comparison of liquid- and gas-phase (flow-cell) reactor for BiOBr catalyst. Reprinted with permission from Reference [15].

### 3.3.2. Carbon Monoxide

Various factors influencing the GDE system performance, such as electrolyte, gas diffusion layer, and pressure, have been tuned for CO production [23,30,32,71,73,74,78,79]. Indeed, this had led to significant advances in terms of the current density for $CO_2RR$. The Kenis research group investigated the effects of the electrolyte composition and carbon support type with Ag and Au catalysts [32,75]. From a comparison of Ag catalyst activities with different KOH concentrations (0.5 to 3.0 M), an improvement was found as the concentration increased [75]. For example, a $j_{CO}$ value as high as 440 mA cm$^{-2}$ was recorded using 3.0 M KOH. Furthermore, in addition to the electrolyte concentration affecting the $CO_2RR$, it was also found that anions significantly influence the $CO_2$ reduction. Hydroxide (OH$^-$) reduced the onset potential more effectively than bicarbonate (HCO$_3{}^-$) and chloride (Cl$^-$). This indicates that the electrolyte plays a key role in the $CO_2RR$ process. The Kenis group presented new insight into $CO_2RR$ at a high pH by carrying out further systematic investigations with Au nanoparticles, supported on poly polymer wrapped multiwall carbon nanotubes (MWNT/PyPBI/Au) [32]. In the performance evaluation, a combination of desirable properties for catalytic activity, such as a small Au particle size, high active surface area, and the high conductivity of multiwall carbon nanotubes, made it possible to obtain an exceptional $j_{CO}$ and overpotential (158 mA cm$^{-2}$ at −0.55 V vs. RHE) with MWNT/PyPBI/Au electrodes compared with commercial Au. A Tafel analysis then was performed on the prepared Au, with various electrolytes with pH values from 6.54 to 13.77 (Figure 13a). The Tafel slopes in this work were similarly obtained between 115 and 133 mV decade$^{-1}$, regardless of the electrolyte type, indicating that the rate-determining step is determined by the $CO_2$ radical formation, which is the first electron transfer process of the $CO_2$ reduction reaction to CO. However, Figure 13b shows that a lower cathodic overpotential could be achieved at a higher pH value, suggesting that the overall $CO_2RR$ process depends on the pH value. That is, the use of an electrolyte with a high pH can be an effective strategy for more facile $CO_2RR$. As another approach for the improved catalytic activity, the collaborative work by the Sargent and Sinton groups reported an Ag-based GDE exhibiting the lowest overpotential for CO (300 mV) at an operating current density of 300 mA cm$^{-2}$, with nearly a 100% selectivity, by increasing the pressure to 7 atm in a highly alkaline condition (7.0 M KOH) (Figure 14). This emphasizes that a microfluidic reactor combining the strategies of higher pressure and an alkaline environment can provide a feasible route for a commercial $CO_2$ conversion system.

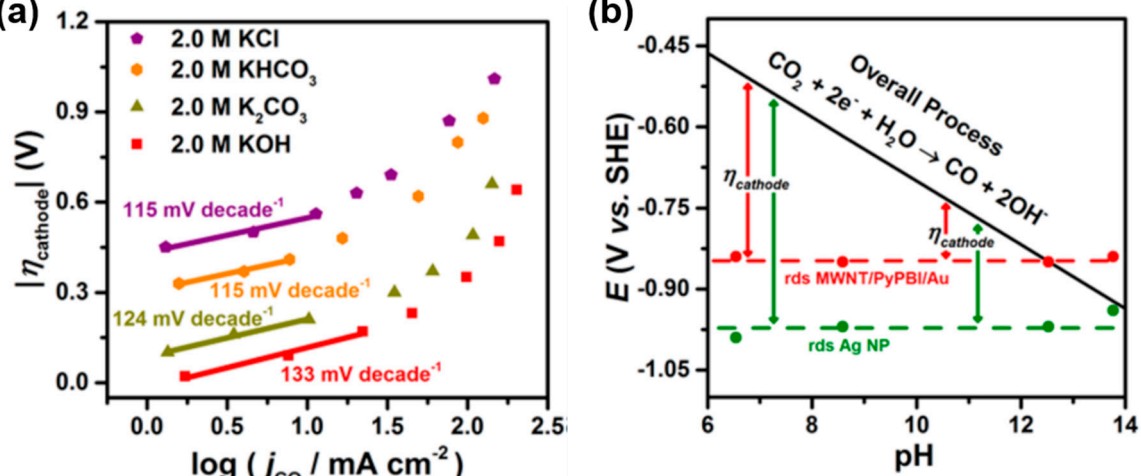

**Figure 13.** The results of electrolyte dependence for the electrochemical $CO_2$ reduction reaction. (**a**) Tafel plots when using various electrolytes (KCl, KHCO$_3$, K$_2$CO$_3$, and KOH). (**b**) Diagram of the relationship between the onset cathodic potential and pH values. Reprinted with permission from the authors of [32]. Copyright (2018) American Chemical Society.

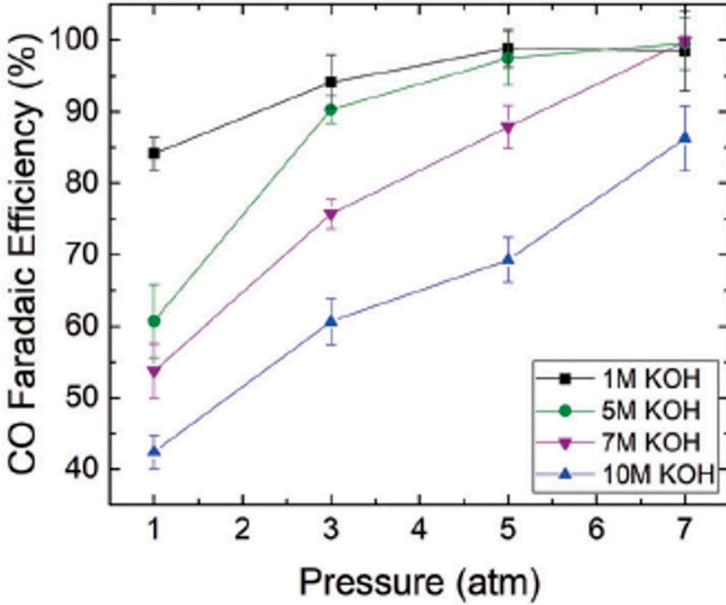

**Figure 14.** The effect of cell pressure on CO Faradaic efficiency and electrolyte concentrations at an operating condition of 300 mA cm$^{-2}$. Reprinted with permission from the authors of [23].

### 3.3.3. Hydrocarbons

Recently, microfluidic reactor systems producing highly value-added hydrocarbon chemicals have attracted considerable interest, leading to significant advances in the overall catalytic performance, whereas research involving MEA systems is lacking regarding hydrocarbon products [14,16–18,60,68,75,83]. In fact, most of the reported studies in this field have been presented in very recent years, despite the fact that the pioneering demonstration of a microfluidic configuration was reported in 1990, with Cu catalysts producing the main products of $CH_4$ and $C_2H_4$ [60]. Because of the complex reaction pathways to release the hydrocarbons as the final product in $CO_2RR$, the improving catalytic properties favoring a reaction for hydrocarbon (e.g., C–C coupling) is a crucial issue for high selectivity. Therefore, many research groups have focused on developing novel catalyst materials by tailoring the structure, composition, and so on, for this purpose.

Figure 15 shows some examples of previously proposed catalysts for GDE [18,68,72]. Zhuang et al. succeeded in multi-carbon alcohol production by engineering Cu catalysts with sulfur atoms. Based on the lessons regarding the role of metal atomic vacancies controlling the reaction intermediates from the liquid-phase research of $CO_2$ reduction, they designed copper sulfide structures for stable surface defects (Figure 15a) [119]. To understand the $C_2$ reaction pathway on the designed $Cu_2S$-Cu-V catalyst, the computational and experimental studies in a H-type cell were employed for a general analysis prior to GDE application. From the results, the DFT studies suggest that subsurface sulfur atoms and Cu atom vacancy defects can promote the reaction to ethanol by suppressing the ethylene evolution. Also, they experimentally observed $C_3H_7OH$ and $C_2H_5OH$ with Faradaic efficiencies of 8% and 15%, respectively, in a $CO_2$-saturated 0.1 M $KHCO_3$ solution. The $Cu_2S$-Cu-V catalysts exhibited a superior activity for alcohol production in a microfluidic reactor cell operated with a KOH electrolyte, which is expected to be more beneficial for driving the ethanol pathway. Specifically, a partial current density of 126 mA cm$^{-2}$ and a 32% selectivity were achieved for the overall multi-carbon alcohols. Moreover, CuAg metal alloy and N-doped graphene quantum dot (NGQD) metal-free catalysts were studied for not only ethylene, but also ethanol production (Figure 15b,c).

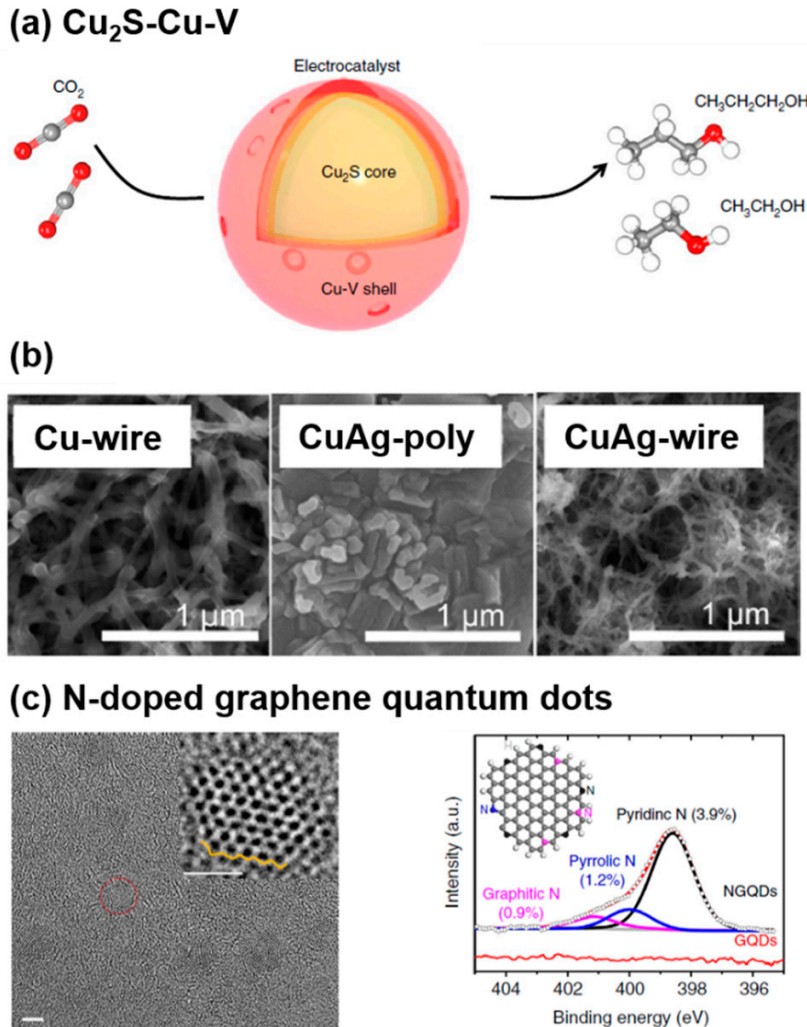

**Figure 15.** Various catalytic materials examples for a gas diffusion electrode in a microfluidic cell to target hydrocarbon production. (**a**) Cu$_2$S-Cu-V core-shell, (**b**) Cu and CuAg with different forms, and (**c**) nitrogen doped graphene quantum dots. Reprinted with permission from References [18,68,72]. Copyright (2018) American Chemical Society for [68].

Sargent and co-workers achieved an outstanding record of ethylene production, with a 70% selectivity at a significantly low applied potential of −0.55 V (vs. RHE) with Cu GDE catalysts [3]. A Tafel analysis suggested that hydroxide concentrations (1.0 M–10.0 M) strongly affect the rate-determining step of CO$_2$RR for C$_2$H$_4$ production on a 100-nm-thick Cu deposited GDE. The DFT results further supported that hydroxide ions impact CO dimerization, which is the rate-determining process for C$_2$ in case of a high pH condition and low potentials. In addition, the thickness of the deposited Cu was tuned from 10 to 1000 nm. Among the prepared samples, the 25-nm-thick Cu samples exhibited an optimized C$_2$H$_4$ selectivity of over 60%. Finally, they noted that the traditional GDE architectures have shown poor stability, typically caused by the flooding phenomena of the gas diffusion layer during electrolysis. To overcome this problem, they proposed a new electrode structure—a graphite/carbon nanoparticles/Cu/polytetrafluoroethylene (PTFE) layer. The new Cu electrode showed an exceptional durability, maintaining about a 70% C$_2$H$_4$ selectivity for 150 hours, while the typical Cu GDE suffered rapid degradation. The highly stable characteristic results from the high hydrophobicity of the PTFE layer, preventing flooding of the gas diffusion layer and passivation by the carbon nanoparticle and graphite layers for Cu catalysts.

## 4. Summary and Outlook

Electrochemical $CO_2$ reduction for value-added carbon products has been developed along various routes for a few decades. The approaches are classified into two categories, liquid- and gas-phase systems. Despite the low $CO_2$ solubility limiting productivity, a tremendous amount of research has been carried out with liquid-phase reactors (H-type cells). With brief reviews on these reactors, major lessons were highlighted for material design and the mechanistic insight for $CO_2RR$. We also reviewed previous research on gas-phase reactors, which are more prominent for practical applications. In particular, we classified the configuration of gas-phase systems, membrane electrode assembly, and microfluidic cells. The representative examples discussed in Section 3 are summarized in Table 2. Researchers have attempted to optimize the various parameters concerning the catalytic electrode, membrane, and reaction environmental conditions, in efforts to enhance the overall performance for both reactors.

(1) MEA systems for $CO_2RR$ are developed more closely to an industrial level, as they are translated from fuel cell technology. Among the reported studies, the MEA reactor performance for CO production with a Sustainion membrane by Dioxide Materials reached an industrially relevant level, particularly given its exceptional durability over six months [1]. This emphasizes the importance of an efficient and robust PEM to achieve a superior catalytic activity in a MEA system. In addition to membrane development, the roles of various factors, such as $CO_2$ feed (gas, humidified, and catholyte), pressure, temperature, and buffer electrolytes in the MEA reactor configurations, should be considered for increased optimization, as some perspective documents a well-organized strategy regarding these issues [35,36]. For higher-order carbon products that have larger markets, new efficient catalytic materials on the gas diffusion layer should be developed for MEA systems. In this case, the trapping issue of liquid products in a membrane should be addressed in order to achieve a more promising operation in this configuration. Importantly, for the catalyst's development, it is imperative to study the kinetics by precisely controlling the potentials on the cathode. Because no liquid electrolyte is used in a typical MEA reactor, the potential can be managed via a buffer layer incorporated between the cathode and membrane, or anode and membrane. The buffer layer located between the anode and membrane might be more proper in order to avoid the invasion of the electrolyte into the gas diffusion layer, causing low stability. In this case, however, the *iR* compensation process is highly required because of the potential drops via the membrane.

(2) For microfluidic cells, although the level of the current density for most target products (from $C_1$ to $C_2$) is considerably high, a stability issue remains, mainly caused by flooding of the GDE during electrolysis. According to many reports, a high alkaline condition (typically based on a KOH solution) is favorable, because of the key role of $OH^-$ for the $CO_2$ reduction pathway [14,31]. However, we note that it is significant to avoid the formation of carbonate product (e.g., $K_2CO_3$) as a result of the chemical reaction between $CO_2$ and KOH. Thus, a robust GDE structure for maintaining the hydrophobicity and high catalytic activity is required, by addressing above-mentioned issues for further development. In this cell configuration, system optimization is necessary by manipulating the temperature, pressure, electrolyte, membrane, and so on. Furthermore, the flow field plate, which is a component for the $CO_2$ gas supply into the gas diffusion layer, can also designed with various patterns so as to maximize the overall $CO_2$ conversion rate [120–122].

As a balanced viewpoint of MEA and microfluidic reactors, the development of novel catalytic materials for a gas diffusion electrode is the most important for promoting the $CO_2RR$ for target products. Reviewing previous research in terms of developing a catalyst in a gas-phase system, obviously shows that they have approached the liquid-phase reaction with the similar strategies. It shows that it needs to follow the important lessons from the tremendous studies in the liquid-phase reaction for developing a gas-phase system. In addition, the reactor design should be optimized for a higher catalytic activity and stability, by combining the merits from the MEA and the microfluidic type, respectively. Finally, an electrolyzer stack should be considered in order to address the system scale-up issue for commercialization.

**Table 2.** Summary of research with gas-phase reactor, with important parameters and performance. MEA—membrane electrode assembly.

| Reactor | Catalyst | Product | Electrolyte | $j_{CO2RR, MAX}$ (mA cm$^{-2}$) | Stability | Note | Ref |
|---|---|---|---|---|---|---|---|
| MEA | Sn | formate | None | 51.7 | 48 h | No catholyte, 70 °C | [26] |
| MEA | Sn | formate | None | 148 | 10 h | pH buffer layer | [58] |
| MEA | Sn | formate | KHCO$_3$ with KCl | 110 | 90 m | Sn particle optimization | [86] |
| MEA | Sn | formate | KHCO$_3$ | 13 | 12 h | Sn loading optimization | [91] |
| MEA | Sn | formate | KHCO$_3$ with KCl | 195 | 100 m | Scaling-up test | [64] |
| MEA | Ag | CO | None | 30 | 5.5 h | pH buffer layer | [55] |
| MEA | Ag | CO | K$_2$SO$_4$ with KHCO$_3$ | 280 | 70 m | 24.6 atm and 333 K | [65] |
| MEA | Ag | CO | None | 300 | 6 mon | Sustainion membrane | [87] |
| MEA | Cu | Acetaldehyde, methanol | None | 2.3 | 350 m | The effect of carbon support | [54] |
| MEA | Cu | CH$_4$, C$_2$H$_4$ | K$_2$SO$_4$ | 3.96 | 5 h | Selemion vs. Nafion | [93] |
| Microfluidic | Sn | formate | KCl | 136 | N/A | pH dependence | [27] |
| Microfluidic | Sn | formate | KHCO$_3$ | 145 | N/A | Ultra small SnO$_2$ | [118] |
| Microfluidic | BiBrO | formate | KHCO$_3$ | 200 | N/A | New catalyst | [15] |
| Microfluidic | Au | CO | KOH | 220 | 8 h | Gas diffusion layer optimization, pH dependence | [32] |
| Microfluidic | Cu | C$_2$H$_4$ | KOH | 360 | 4 h | Cu particle optimization | [75] |
| Microfluidic | Cu | CH4, C2H4 | KOH | 404 | N/A | First microfluidic cell | [60] |
| Microfluidic | Cu$_2$S | ethanol, propanol | KOH | 350 | 150 m | New catalyst | [18] |
| Microfluidic | CuAg | C$_2$H$_4$ | KOH | 275 | N/A | CuAg alloy catalyst | [68] |
| Microfluidic | NGQD | CO | KOH | 155 | N/A | New catalyst | [72] |
| Microfluidic | Cu | C$_2$H$_4$ | KOH | 600 | 150 h | New GDE structure | [14] |

**Author Contributions:** J.T.S. and J.O. conceived the main content of the review paper. J.T.S. and H.S. mainly discussed and wrote the content for gas- and liquid-phase CO$_2$ reduction research, respectively. B.K. assisted in the collection of previous studies and in the discussion for the review paper. All of the authors discussed the review and assisted with manuscript preparation.

**Acknowledgments:** This work was supported by a grant from the Korea CCS R&D Center (Korea CCS 2020 Project) (KCRC-2014M1A8A1049303) and the Creative Materials Discovery Program (NRF-2017M3D1A1040692), funded by the Korean government (Ministry of Science and ICT). The authors also acknowledge the support of a KAIST GCORE (Global Center for Open Research with Enterprise) grant funded by the Ministry of Science and ICT (N11190005).

**Conflicts of Interest:** The authors declare no conflict of interest.

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
