# Peer review of "Towards Higher Rate Electrochemical CO2 Conversion: From Liquid-Phase to Gas-Phase Systems"

_catalysts, doi:10.3390/catal9030224_

Round 1

Reviewer 1 Report

This manuscript by Song et al. reviews the advance of CO2RR from the point view of reactor/cell, which brings the readers’ attention to practical applications of this important reaction. This is a very interesting topic and so far only few such reviews with emphasize on the reactor have been published (e.g., Acc. Chem. Res. 2018, 51, 910; Current Opinion in Green and Sustainable Chemistry 2019, 10.1016/j.cogsc.2019.01.005). Nevertheless, I think this review will help guide the researchers in this field and could be published after addressing the following critical concerns.

1. Section 2 discussed catalyst structure as well as the reaction conditions in a liquid-phase reactor. This is well presented. However, Section 3 seems to have mixed the role of configurations of different cells with sometimes also the catalyst structure. The authors should have more clear discussions on the two different aspects. I would suggest the authors to leave the structure and performance only regarding to gas-phase CO2RR in this section, while discussing the general performance in section 2.

2. The authors should pay more attention to the configuration of the MEA reactor. As a translation from fuel cell and water electrolyzer, the MEA reactor might be more close to industrial CO2RR. Its typical configuration is shown in Figure 7a. But the actual configurations vary from group to group, such as an exactly sandwich-like MEA structure as it is in the fuel cell (e.g., ref 26), with electrolytes in between membrane/cathode (e.g., refs 51,48) or membrane/anode (Energy Technol. 2017, 5, 922; Nanotechnology 2018, 29, 014001). Please also think about the role of feeds in these configurations. For sure, gas CO2 is also fed to the GDE, but how does the performance change when using KOH or KHCO3 as the electrolyte in between membrane/cathode? The authors should also discuss the anode side where water vapor (usually with an inert gas), pure liquid water and KOH are fed in the literature. The use of membrane, especially the newly-developed bipolar membrane, (J. Electrochem. Soc., 2019, 166, F34; J. CO₂. Utilization 2018, 23, 152) is also a key aspect. Please think about their different roles in these MEA cells and give some discussion on these important parameters. I do believe these efforts would consolidate this review very well.

3. Section 3.2.3 has to include some new references, since there are actually more relevant reports regarding to selective hydrocarbon/alcohol production in a MEA reactor besides ref47, e.g. Energy Technol. 2017, 5, 922; Nanotechnology 2018, 29, 014001; J. Energy Chem. 2014, 23, 694; Electrochem. Commun. 2015, 50, 64.

4. An important issue: How can the potential be controlled in the MEA reactors where no liquid electrolyte is used? How can a reference electrode be built in the MEA? The precise potential of the cathode instead of only the cell voltage is key to study the reaction mechanisms by in situ techniques, which would be an important direction, since the thermodynamics and kinetics of the CO2RR within flow reactors are fundamentally different to those found in an H-cell environment.

5. The authors should provide a balanced viewpoint on MEA and microfluidic reactors. Can they have a perspective on the development of the two kind of reactors?

6. Although this review focuses on the low-temperature gas phase CO2RR, high temperature CO2RR using oxygen ion or proton conducting solid electrolyte might need to be briefly introduced. The co-electrolysis of CO2 and water in the presence of proton conducting solid electrolyte does produce hydrocarbons with a very high rate.

7. Please correct the caption of Figure 15.

Author Response

We really appreciate the Reviewer #1 for important comments to improve our manuscripts. A point-by-point response by us is uploaded as PDF file.

Reviewer 2 Report

In the present work, the authors propose a further study of gas phase system for CO2 reduction, analysing some previous approaches from liquid-phase to gas-phase systems. In this review, they highlight the major challenges for developing new CO2 conversion systems, leading to evidence different parameters that can affect the electrocatalytic performance toward CO2 reduction reaction.  Among all several parameters, they focused their attention on the low solubility of CO2 in an aqueous solution, how the pH of the electrolyte, the morphology of catalyst, grain boundary can influence catalytic performance.  The adequacy, coherence and completeness of this review work are ensured. However, some changes can be done to better explain the different topics.

The introduction should be written carefully to better organize the explanation of the limits offered by all systems previously investigated for the CO2 conversion, and to have a clearer definition of main aim of work.  The quality of figure 1 should be improved and the citation of required permission should be better specified.  Moreover, in the last paragraph, where the authors declare that "many researchers in this field have recently promoted the development of gas-phase reactors", several reference must be added.

In the chapter 2 The sentence:” a brief review of many factors (structuring, control of active sites, electrolysis environments) contributing to higher catalytic activity, as depicted in Figure 6”, is not quite clear. This concept should be better explained. For example, what concerns the structuring? How could the control of active sites be employed?

In the chapter 3.2.2 the last paragraph of the chapter seems to be focused more on the company than the membrane itself. For this reason it should  be re-written in order to clarify and highlight the good quality and performance of the membrane.

More in general, the permission required for reprinting the figure should be declared in a different way, accordingly to the editor and or journal requirements.

Author Response

We really appreciate the Reviewer #2 for important comments to improve our manuscripts. A point-by-point response by us is uploaded as PDF file.

Reviewer 3 Report

The paper describes the recent works related to CO2 reduction with main focus on differences of liquid vs. gas phase systems. This paper is well writen and in my opinion will present interesting an interesting data for the researchers in the field.

A disadvantage of the paper is the lack of description of the electrodes other than metallic ones. There is an important trend of using molecular catalysts such as for instance porphyrins 10.1021/acs.jpcc.5b10763 or 10.1002/cctc.201701045 for the cathodes.

Minor typos that I have found:

line 130: C2 -> C2 (2 in subscript)

line 139: the Zhang group -> Zhang group

Reference 19 concerns CO reduction rather than CO2 - it should be clarified in text.

Author Response

We really appreciate the Reviewer #3 for important comments to improve our manuscripts. A point-by-point response by us is uploaded as PDF file.

Round 2

Reviewer 1 Report

The authors have addressed my comments and significantly improved the manuscript. I believe it will be a good reference for researchers in this field. I am happy to recommend its publication in Catalysts in the current form.